

# Chiral anomaly trapped in Weyl metals:
# Nonequilibrium valley polarization at zero magnetic field

Pablo M. Perez-Piskunow[1][*], Nicandro Bovenzi[2][†],
Anton R. Akhmerov[3][‡] and Maxim Breitkreiz[4][○]

**1** Catalan Institute of Nanoscience and Nanotechnology (ICN2), CSIC and BIST,
Campus UAB, Bellaterra, 08193 Barcelona, Spain
**2** Instituut-Lorentz, Universiteit Leiden, P.O. Box 9506, 2300 RA Leiden, The Netherlands
**3** Kavli Institute of Nanoscience, Delft University of Technology,
P.O. Box 4056, 2600 GA Delft, The Netherlands
**4** Dahlem Center for Complex Quantum Systems and Fachbereich Physik,
Freie Universität Berlin, 14195 Berlin, Germany

## Abstract

In Weyl semimetals, the application of parallel electric and magnetic fields leads to valley polarization—an occupation disbalance of valleys of opposite chirality—a direct consequence of the chiral anomaly. In this work, we present numerical tools to explore such nonequilibrium effects in spatially confined three-dimensional systems with a variable disorder potential, giving exact solutions to leading order in the disorder potential and the applied electric field. Application to a Weyl-metal slab shows that valley polarization also occurs without an external magnetic field as an effect of chiral anomaly "trapping": Spatial confinement produces chiral bulk states, which enable the valley polarization in a similar way as the chiral states induced by a magnetic field. Despite its finite-size origin, the valley polarization can persist up to macroscopic length scales if the disorder potential is sufficiently long ranged, so that direct inter-valley scattering is suppressed and the relaxation then goes via the Fermi-arc surface states.



# 1 Introduction

The most famous effect associated with Weyl Fermions is the chiral anomaly [1,2]—magnetic-field induced chiral states moving parallel or antiparallel to the field, depending on the chirality of the Weyl Fermion. In Weyl semimetals [3,4] the two chiralities occur pairwise, ensuring an equal number of forward- and backward-propagating states, and the chiralities are connected by Fermi-arc surface states. Existing Weyl-semimetal materials typically have a small but finite Fermi momentum $k_F$ measured from the Weyl node and a much larger momentum-space separation $\Delta k$ of valleys that host the opposite chiralities.

The valley degree of freedom [5] plays a central role in the transport behavior of Weyl semimetals. Parallel electric and magnetic fields produce a difference in the non-equilibrium occupation of valleys [3,6,7]. A direct consequence of this valley polarization is an enhanced conductivity parallel to the magnetic field due to the polarization-enhanced occupation disbalance of countermoving chiral states [8–13]. Experimental observations, although obscured by the competing current-jetting effect [14], support the general feature of a chiral-anomaly enhancement of the conductivity [15–17]. Other manifestations of the valley degree of freedom are found in nonlocal transport measurements [6,18] and in the photogalvanic response [19,20].

Crucial in understanding "valleytronic" transport is to explore the effect of disorder and the finite size of the crystal, which are two unaviodable properties of real materials. Disorder plays a subtle role if the Fermi level lies at the Weyl nodes, where it may or may not destroy the ideal semimetal phase by inducing a finite density of states [21–24] or, for finite inter-valley scattering, drive the system into an insulating phase [25]. At finite chemical potentials, well-separated Weyl nodes, and a weak disorder potential the Weyl-semimetal phase has proven to be robust, allowing for a perturbative treatment of disorder, which will be employed in this work.

Finite-size effects in meso- and macroscopic Weyl semimetals (crystal dimensions much larger than the lattice constant), have also been explored with the focus on the role of topological Fermi-arc surface states [26–36]. Peculiarities are rooted in the specifics of the momentum-space structure of Fermi arcs connecting valleys of opposite chirality and their unidirectional motion at a single surface, see Fig. 1. Separation of countermoving Fermi arcs to opposite surfaces explains their relevance at large systems sizes, most prominently in the intrinsic anomalous Hall effect [34,37–39]. The relevance of finite-size effects for the valley degree of freedom, on the other hand, is much less obvious, since the valleys consist of extended bulk states, lacking

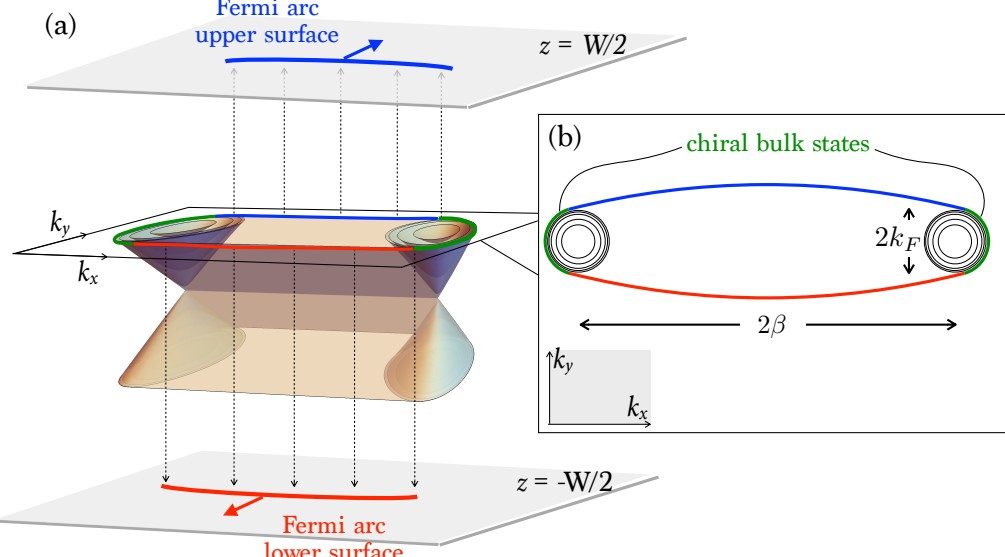

Figure 1: (a) Schematic picture of the considered Weyl-slab model in a mixed momentum-/real-space illustration. The plot shows the slab spectrum as a function of in-plane momenta. Top (blue) and bottom (red) surface states are indicated with their position and propagation direction in real space. Chiral bulk states connecting surface states of both surfaces are depicted in green. (b) Top-view on equi-energy contours of the slab spectrum of (a).

the spatial separation.

In this work, we show that in a disordered Weyl semimetal slab valley polarization can be induced without external magnetic fields as a finite-size effect at mesoscopic slab widths, possibly extending to even larger sizes. Crucial turn out to be confinement-induced chiral bulk states [33, 36]: At zero magnetic field and a finite Fermi momentum $k_F$ there is a residual density of chiral bulk states, which must remain to reconnect the two Fermi-arc surface states as shown in Fig. 1. The density of chiral bulk states of a single valley, relative to the density of magnetic-field induced chiral states, is $k_F/l_B^{-2}W$, where $l_B = \sqrt{\hbar/eB} \approx 26\,\mathrm{nm}\sqrt{1/B[\mathrm{T}]}$ is the magnetic length, and $W$ the width of the slab. Taking an experimentally realistic value of $k_F = 0.01\,\text{Å}^{-1}$, the density of anomalous chiral states is larger than that of magnetic-field induced ones for $W \lesssim 100\,\mathrm{nm}\,/B[\mathrm{T}]$. At a mesoscopic width $W \sim 100\,\mathrm{nm}$, the effect of anomalous chiral states is thus comparable to that of the field-induced chiral states at $B \sim 1\,\mathrm{T}$, which relevance is commonly accepted and experimentally supported [15–17].

We find that the confinement-induced valley polarization and the presence of surface states can lead to conductivity enhancements by several orders of magnitude, compared to that of the infinite system. This conductivity enhancement is suppressed with increasing width $W$ as $1/W$, simply due to the decreasing density of confinement-induced chiral states. The valley polarization, however, turns out to remain unsupressed up to widths set by the probability of direct inter-valley scattering, which in case of Gaussian-type disorder poteials is exponentially enhanced $\sim \exp\left[(\xi\Delta k)^2\right]$.

To reveal this effect, we develop a two-part numerical approach combining the full quantum mechanical calculations of a multiband slab dispersion and wavefunctions, with a numerical solution for the non-equilibrium corrections to the density matrix [40]. The resulting non-equilibrium density matrix is exact to leading order in the disorder potential and the applied external electric field.

The paper is organized as follows. In Section II we start with a derivation of transport

equations for multiband systems with a large number of bands and discuss its validity regime. In Section III we introduce the model of the Weyl slab, the impurity potential, and calculate the scattering rate. In Section IV we present the transport results obtained from solving the transport equations numerically, which we discuss in Section V by comparing with simplified analytic calculations. We conclude in Section VI.

## 2 Quantum transport approach

In the first part of this section we recapitulate the general transport formalism in the presence of weak disorder following Kohn and Luttinger [41]. This is necessary to identify the validity regime of this formalism when applied to a spatially confined system, which we do in the second part.

### 2.1 General quantum transport approach

We separate a general single-particle Hamiltonian into the free-particle part $H_0$, the additional weak scattering potential $V$, and a time-dependent electric-field term $e\boldsymbol{E} \cdot \boldsymbol{r} \, e^{st}$ with the position operator $\boldsymbol{r}$ and an adiabatic time dependence $e^{st}$ with $s \to 0^+$,

$$H = H_0 + V + e^{st} e\boldsymbol{E} \cdot \boldsymbol{r} \, . \tag{1}$$

The scattering is due to a random configuration of impurities with a vanishing impurity-averaged potential $\langle\langle V \rangle\rangle = 0$.

We make the ansatz for the full density matrix

$$\rho = p + g e^{st} \, , \tag{2}$$

where $p$ is the equilibrium density matrix

$$p = \frac{e^{-\beta(H_0 + V)}}{\operatorname{Tr} e^{-\beta(H_0 + V)}} \tag{3}$$

and $g$ the non-equilibrium correction. Inserting into the von Neumann equation for the density matrix,

$$i\partial_t \rho = [H, \rho], \tag{4}$$

and expanding to first order in $E$ we obtain

$$i s \, g = [e\boldsymbol{E} \cdot \boldsymbol{r}, p] + [H_0 + V, g]. \tag{5}$$

The following analysis consists in expanding (5) in powers of $V$. We write (5) in terms of its matrix elements in the basis of $H_0$ eigenstates $|\boldsymbol{\kappa}\rangle$, where $\boldsymbol{\kappa}$ combines the quantum numbers. Off-diagonal and diagonal elements read, respectively,

$$(E_{\boldsymbol{\kappa}} - E_{\boldsymbol{\kappa}'} - is) g_{\boldsymbol{\kappa}\boldsymbol{\kappa}'} = (g_{\boldsymbol{\kappa}} - g_{\boldsymbol{\kappa}'}) V_{\boldsymbol{\kappa}\boldsymbol{\kappa}'} + C_{\boldsymbol{\kappa}\boldsymbol{\kappa}'} + \sum_{\boldsymbol{\kappa}'' \neq \boldsymbol{\kappa}', \boldsymbol{\kappa}} (g_{\boldsymbol{\kappa}\boldsymbol{\kappa}''} V_{\boldsymbol{\kappa}''\boldsymbol{\kappa}'} - V_{\boldsymbol{\kappa}\boldsymbol{\kappa}''} g_{\boldsymbol{\kappa}''\boldsymbol{\kappa}'}) \, , \tag{6}$$

$$-is g_{\boldsymbol{\kappa}} = C_{\boldsymbol{\kappa}} + \sum_{\boldsymbol{\kappa}' \neq \boldsymbol{\kappa}} (g_{\boldsymbol{\kappa}\boldsymbol{\kappa}'} V_{\boldsymbol{\kappa}'\boldsymbol{\kappa}} - V_{\boldsymbol{\kappa}\boldsymbol{\kappa}'} g_{\boldsymbol{\kappa}'\boldsymbol{\kappa}}) \, , \tag{7}$$

where the notation is $A_{\boldsymbol{\kappa}\boldsymbol{\kappa}'} = \langle \boldsymbol{\kappa} | A | \boldsymbol{\kappa}' \rangle$, $A_{\boldsymbol{\kappa}} = A_{\boldsymbol{\kappa}\boldsymbol{\kappa}}$, $E_{\boldsymbol{\kappa}} = \langle \boldsymbol{\kappa} | H_0 | \boldsymbol{\kappa} \rangle$. The field-dependent term

$$C_{\boldsymbol{\kappa}\boldsymbol{\kappa}'} = e\boldsymbol{E} \cdot [\boldsymbol{r}, p]_{\boldsymbol{\kappa}\boldsymbol{\kappa}'} \tag{8}$$

expands in powers of $V$ starting with the zeroth order,

$$C^{(0)}_{\kappa\kappa'} = e\boldsymbol{E}\cdot\boldsymbol{r}_{\kappa\kappa'}\left[n_F(E_\kappa)-n_F(E_{\kappa'})\right], \tag{9}$$

where $n_F(E)$ is the Fermi distribution. From (6) and (7) we see that the leading order of the off-diagonals of $g$ are of order $V^{-1}$, while the diagonals are of order $V^{-2}$. To leading order, the latter two terms in (6) can thus be neglected, leading to

$$g_{\kappa\kappa'} = \frac{(g_\kappa - g_{\kappa'})V_{\kappa\kappa'}}{E_\kappa - E_{\kappa'} - is}\ . \tag{10}$$

Inserting into (7), taking the adiabatic limit $s\to 0^+$, and applying disorder averaging $\langle\!\langle\ldots\rangle\!\rangle$ we obtain

$$C^{(0)}_\kappa = i\,2\pi\sum_{\kappa''\neq\kappa}\delta(E_\kappa - E_{\kappa'})\left\langle\!\left\langle|V_{\kappa\kappa'}|^2\right\rangle\!\right\rangle(g_{\kappa'}-g_\kappa). \tag{11}$$

If the electric field points in a direction in which the system is infinite, let it be $x$ and $y$, the eigenstates can be chosen as momentum $\boldsymbol{k}=(k_x,k_y)$ eigenstates. The field term (9) becomes

$$C^{(0)}_\kappa = i\,e\boldsymbol{E}\cdot\boldsymbol{v}_\kappa\, n'_F(E_\kappa), \tag{12}$$

where $\boldsymbol{v}_\kappa = \partial_\kappa E_\kappa$ is the velocity.

Making the ansatz

$$g_\kappa = -e\boldsymbol{E}\cdot\boldsymbol{\Lambda}_\kappa\, n'_F(E_\kappa) \tag{13}$$

(11) simplifies to

$$n'_F(E_\kappa)\boldsymbol{v}_\kappa = 2\pi\sum_{\kappa'\neq\kappa}n'_F(E_\kappa)\delta(E_\kappa - E_{\kappa'})\left\langle\!\left\langle|V_{\kappa\kappa'}|^2\right\rangle\!\right\rangle(\boldsymbol{\Lambda}_\kappa - \boldsymbol{\Lambda}_{\kappa'}), \tag{14}$$

known as Boltzmann equation (BE), to be solved with respect to the vector-valued state-resolved transport mean free paths $\boldsymbol{\Lambda}_\kappa$, which we will refer to as *transport length* in short. The average magnitude of the transport length scales with the strength of the impurity potential as $V^{-2}$.

Note that since the summation operator acting on $\boldsymbol{\Lambda}_\kappa$ in (14) has an eigenvalue zero for a $\kappa$ independent vector, the solution is generally determined up to a constant

$$\boldsymbol{c} = \frac{\sum_\kappa n'_F(E_\kappa)\boldsymbol{\Lambda}_\kappa}{\sum_\kappa n'_F(E_\kappa)}\ . \tag{15}$$

Particle conservation however requires $\sum_\kappa g_\kappa = 0$, which fixes the constant to $\boldsymbol{c}=0$.

The current-density expectation value reads

$$\boldsymbol{j} = -e\frac{1}{\mathcal{V}}\sum_\kappa \boldsymbol{v}_\kappa\, g_\kappa, \tag{16}$$

where $\mathcal{V}$ is the system volume. The conductivity tensor $\sigma$, defined as $\boldsymbol{j} = \sigma\boldsymbol{E}$, becomes, using (13),

$$\sigma = e^2\frac{1}{\mathcal{V}}\sum_\kappa n'_F(E_\kappa)\boldsymbol{v}_\kappa \otimes \boldsymbol{\Lambda}_\kappa. \tag{17}$$

Note that the BE (14) is exact in the weak-disorder limit, giving a conductivity that scales with the squared inverse strength of the disorder potential. Leading corrections, which will not be considered here, are of zeroth order in the impurity potential, they include, e.g., the anomalous Hall effect.

## 2.2 Application to a slab model

We now discuss the validity regime of (14) when applied to a slab model. We consider a system that is infinite in two spatial directions, $x$ and $y$ (as specified above), and confined in direction $z$ to $-W/2 < z < W/2$. The slab energy eigenspace $\boldsymbol{\kappa} = (\boldsymbol{k}, b)$, where $\boldsymbol{k}$ is the in-plane momentum, has the particularity that the number of bands (band index $b$) is potentially very large, scaling with the width $W$ of the system. Since the BE that we have just derived relies only on considering the leading order in the scattering potential $V$, it can still be applied to the slab, provided that $V$ can be taken to be arbitrary small. For the slab, a problem arises if we want to consider such a large width $W$ that the effect of boundaries becomes smaller than that of the impurity scattering, which can invalidate the expansion in powers of $V$. We now examine when exactly the width becomes "too large" in that sense.

The large width $W$ enters our above formalism through the position matrix elements in $C_{\boldsymbol{\kappa}\boldsymbol{\kappa}'}$, see Eq. (8). Let us thus repeat the above steps without neglecting higher $V$ orders in $C$ since they might still be large due to $W$. In this case the BE obtains an extra term on the right-hand side (rhs), so that the extended BE reads

$$C_{\boldsymbol{\kappa}} = \sum_{\boldsymbol{\kappa}' \neq \boldsymbol{\kappa}} \delta(E_{\boldsymbol{\kappa}} - E_{\boldsymbol{\kappa}'})|V_{\boldsymbol{\kappa}\boldsymbol{\kappa}'}|^2 (g_{\boldsymbol{\kappa}'} - g_{\boldsymbol{\kappa}}) + \sum_{\boldsymbol{\kappa}' \neq \boldsymbol{\kappa}} \delta(E_{\boldsymbol{\kappa}} - E_{\boldsymbol{\kappa}'})(C_{\boldsymbol{\kappa}\boldsymbol{\kappa}'} V_{\boldsymbol{\kappa}'\boldsymbol{\kappa}} - V_{\boldsymbol{\kappa}\boldsymbol{\kappa}'} C_{\boldsymbol{\kappa}'\boldsymbol{\kappa}}). \tag{18}$$

Now expanding $C$ in powers of $V$ the rhs term with $C^{(0)}$ vanishes upon impurity averaging since the mean potential due to impurities is zero. We thus consider the next order term,

$$C_{\boldsymbol{\kappa}\boldsymbol{\kappa}'}^{(1)} = e\boldsymbol{E} \cdot \sum_{\boldsymbol{\kappa}''} \left[ \boldsymbol{r}_{\boldsymbol{\kappa}\boldsymbol{\kappa}''} V_{\boldsymbol{\kappa}''\boldsymbol{\kappa}'} \frac{n_F(E_{\boldsymbol{\kappa}''}) - n_F(E_{\boldsymbol{\kappa}'})}{E_{\boldsymbol{\kappa}''} - E_{\boldsymbol{\kappa}'}} - \boldsymbol{r}_{\boldsymbol{\kappa}''\boldsymbol{\kappa}'} V_{\boldsymbol{\kappa}\boldsymbol{\kappa}''} \frac{n_F(E_{\boldsymbol{\kappa}}) - n_F(E_{\boldsymbol{\kappa}''})}{E_{\boldsymbol{\kappa}} - E_{\boldsymbol{\kappa}''}} \right].$$

While it also vanishes upon averaging on the left-hand side (lhs) of (18), the new term on the rhs becomes

$$e\boldsymbol{E} \cdot \sum_{\boldsymbol{\kappa}' \neq \boldsymbol{\kappa}} \sum_{\boldsymbol{\kappa}''} \delta(E_{\boldsymbol{\kappa}} - E_{\boldsymbol{\kappa}'}) \frac{n_F(E_{\boldsymbol{\kappa}}) - n_F(E_{\boldsymbol{\kappa}''})}{E_{\boldsymbol{\kappa}} - E_{\boldsymbol{\kappa}''}} \left[ V_{\boldsymbol{\kappa}''\boldsymbol{\kappa}'} V_{\boldsymbol{\kappa}'\boldsymbol{\kappa}} \boldsymbol{r}_{\boldsymbol{\kappa}\boldsymbol{\kappa}''} + V_{\boldsymbol{\kappa}\boldsymbol{\kappa}'} V_{\boldsymbol{\kappa}\boldsymbol{\kappa}''} \boldsymbol{r}_{\boldsymbol{\kappa}''\boldsymbol{\kappa}'} \right].$$

In this sum there are terms that are proportional to $|V_{\boldsymbol{\kappa}\boldsymbol{\kappa}'}|^2$, which certainly do not vanish upon impurity averaging. Compared to the first term on the rhs of (18) with the general ansatz (13), the new term is generally smaller if the position matrix elements are smaller than the typical values of the transport length, let us denote them by $\bar{\Lambda}$. This correction can thus be neglected only if the width is much smaller,

$$W \ll \bar{\Lambda}. \tag{19}$$

Higher order terms due to the expansion of $C$ are of order $W$ times a higher power of $V$ and thus give even smaller corrections.

Summarizing section III, Eq. (19) characterizes the validity regime of the BE (14) if applied to the slab. In words, one is allowed to consider impurity scattering as a weak perturbation to a free propagation in the slab treated as a two-dimensional multiband system as long as the mean free path is much larger than the width.

# 3 Weyl-semimetal slab model

We consider a minimal lattice model of a Weyl semimetal [42],

$$H_0(\boldsymbol{\kappa}) = t\sigma_x \sin k_z + t\sigma_y \sin k_y + m_{\boldsymbol{\kappa}}\sigma_z, \tag{20}$$

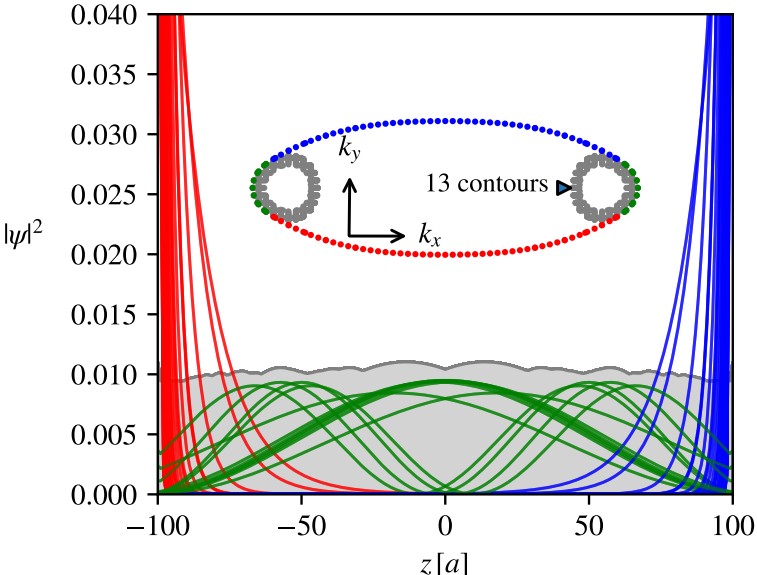

Figure 2: Position-$z$-resolved probability density $|\psi|^2 = |\langle z|\psi\rangle|^2$ of Fermi-level states. Colors indicate surface states at the lower boundary (red) and upper boundary (blue), the confinement-induced chiral bulk states (green), and normal bulk states (gray). The inset shows the position of the states in momentum space. Other parameters are $E_F = 0.3\,t$, $W = 201$, $\mu_b = 0.3\,t$.

where $m_{\kappa} = t(2 + \cos\beta - \cos k_x - \cos k_y - \cos k_z)$ and $\sigma_i$ are pseudospin Pauli matrices (corresponding to an arbitrary degree of freedom), $t$ is the hopping amplitude, and the lattice constant is set to unity. The two Weyl nodes are placed at $\boldsymbol{k} = \pm\boldsymbol{\beta}$, where $\boldsymbol{\beta} = \beta\hat{\boldsymbol{x}}$ corresponds to a time-reversal breaking magnetization. We consider a "good" Weyl semimetal with a cone separation $\beta \sim 1$.

The Hamiltonian of the slab is given by the lattice Hamiltonian (20) but for a finite number $W$ of sites in the $z$ direction. Transformation into the site basis in the $z$ direction replaces $\cos k_z \to (\delta_{i,j+1} + \delta_{i,j-1})/2$, where $i = 0, 1, \ldots, (W-1)$ is the site number, corresponding to the discrete position in $z$,

$$z \equiv i - \frac{W-1}{2}, \tag{21}$$

in units of the lattice constant which is set to one. We furthermore add a boundary potential $\mu_b$ at the surface layers of the slab, which main effect is to bend the Fermi-arc surface states. We label the eigenstates by $\boldsymbol{\kappa} = (\boldsymbol{k}, b)$ where $\boldsymbol{k} = (k_x, k_y)$ are the continuous in-plane momenta and $b$ denotes the $2W$ modes at each value of $\boldsymbol{k}$.

The eigenstates $|\psi_{\boldsymbol{\kappa}}\rangle$ and eigenenergies $E_{\boldsymbol{\kappa}}$ of the slab are obtained from exact diagonalization of the Hamiltonian at a fixed in-plane momentum $\boldsymbol{k}$ using standard methods of numerical diagonalization [40]. For our transport considerations we need to take into account all Fermi-level states, which are continuous contours in the space of the in-plane momentum $\boldsymbol{k}$. We numerically [40] determine the contours by means of the *marching squares* algorithm [43], whereby the contours are discretized. The precision level of the discretization is improved until full convergence of the results. Figure 2 illustrates typical results of numerical diagonalization giving the Fermi-level contours (inset, see also Fig. 1) and the wavefunction probability density $|\psi|^2 \equiv |\langle z|\psi\rangle|^2$.

Most bulk states form closed contours located at one of the valleys. Additionally there is the special contour that connects the valleys by wrapping around them. In between it consists

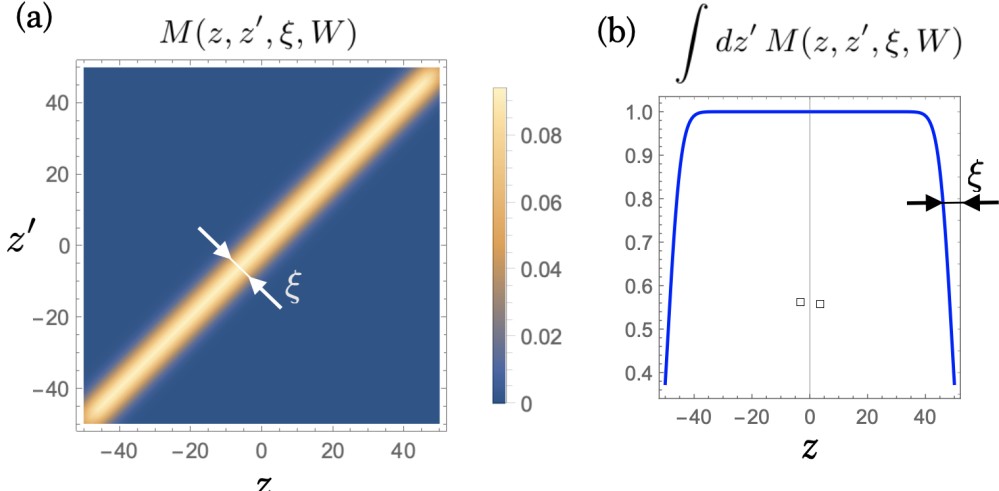

Figure 3: Plot of the function $M(z, z', \xi, W)$ (a) and $\int dz'\, M(z, z', \xi, W)$ (b) at $\xi = 3$ and $W = 100$. The width of the diagonal peak in (a) and the region of reduced weight in (b) are both set by $\xi$.

of surface states but at places where the contour touches the bulk contours, the states are delocalized — we call them chiral bulk states, since they are unidirectional, moving parallel or antiparallel to the intrinsic magnetization (here $x$ direction) depending on the valley. While the number of bulk contours $\sim k_F W/\pi$ increases with the width, there is always only a single contour that contains the surface and chiral bulk states.

## 4 Scattering

### 4.1 Disorder potential

We model the disorder by static Gaussian potentials,

$$V = \sum_\alpha U_\alpha \phi(\boldsymbol{r} - \boldsymbol{r}_\alpha), \qquad \phi(\boldsymbol{r}) = e^{-|\boldsymbol{r}|^2/2\xi^2}, \qquad (22)$$

where the sum runs over the Gaussian's with a characteristic width $\xi$, random and uncorrelated potential magnitudes $U_\alpha \in [-\delta, \delta]$, and random positions $\boldsymbol{r}_\alpha$.

The disorder potential enters the BE (14) in the form of the scattering rate between two energy eigenstates $Q(\boldsymbol{\kappa}, \boldsymbol{\kappa}') = 2\pi\delta(E_{\boldsymbol{\kappa}} - E_F)q(\boldsymbol{\kappa}, \boldsymbol{\kappa}')$, where

$$q(\boldsymbol{\kappa}, \boldsymbol{\kappa}') = \left\langle\!\!\left\langle |\langle \psi_{\boldsymbol{\kappa}}|V|\psi_{\boldsymbol{\kappa}'}\rangle|^2 \right\rangle\!\!\right\rangle \qquad (23)$$

and $\langle\!\langle \ldots \rangle\!\rangle$ denotes disorder average.

In the slab model, as compared to a translation-invariant system, the scattering rate is not a simple Gaussian as a function of the momentum difference. Inserting the impurity potential and averaging over the disorder configurations within the slab we obtain

$$q(\boldsymbol{\kappa}, \boldsymbol{\kappa}') = \frac{(2\pi\xi^2)^3 \delta^2 n_i}{3 L_x L_y} e^{-\xi^2 (\boldsymbol{k} - \boldsymbol{k}')^2/2} \sum_{z, z'} M(z, z', \xi, W) \psi_{\boldsymbol{\kappa}'}^\dagger(z)\psi_{\boldsymbol{\kappa}}(z)\psi_{\boldsymbol{\kappa}}^\dagger(z')\psi_{\boldsymbol{\kappa}'}(z'), \qquad (24)$$

where $\psi_{\boldsymbol{\kappa}}(z) = \langle z|\psi_{\boldsymbol{\kappa}}\rangle$ and $n_i$ is the impurity concentration. A detailed derivation can be found

in the Appendix. The function $M(z, z', \xi, W)$ is given by

$$M(z, z', \xi, W) = \frac{e^{-(z-z')^2/4\xi^2}}{4\sqrt{\pi}\xi} \left[ \text{erf}\left( \frac{W + z + z'}{2\xi} \right) + \text{erf}\left( \frac{W - z - z'}{2\xi} \right) \right],$$

where the error function is defined as $\text{erf}(x) = (2/\sqrt{\pi}) \int_0^x e^{-t^2} dt$. As illustrated in Fig. 3, $M(z, z', \xi, W)$ is mainly the $(z-z')$ dependent part of the Gaussian impurity potential, which magnitude however reduces by approximately a factor of 2 at the slab surfaces, where the possible impurity positions obviously fill only half of the space. Another effect of the finite size is that the $z$ dependence of the wavefunctions is not plane-wave like, hence the sum in (24) does not reduce to a Fourier transformation. In particular, note that the wavefunction factor in (24) strongly suppresses the scattering rate between surface states of opposite surfaces when $\xi$ and the penetration depth of the surface states are both much smaller than $W$ due to a vanishing overlap between the surface states.

In the limit $\xi \ll 1$ one obtains $M(z, z', \xi, W) = \delta(z - z')$, in which case

$$q(\kappa, \kappa') \xrightarrow{\xi \ll 1} \frac{(2\pi\xi^2)^3 \delta^2 n_i}{3 L_x L_y} e^{-\xi^2(k-k')^2} \sum_z |\psi_{\kappa'}(z)|^2 |\psi_\kappa(z)|^2.$$

Despite the rather complex form, the impurity scattering is fully determined by two parameters — the real-space impurity width $\xi$ and the overall impurity strength (set by $\delta^2 n_i$), the latter will be in the following quantified by the average mean free path $l$, defined below. The impurity width $\xi$ essentially sets the momentum-space range of most scattering processes to $|k - k'| \lesssim \xi^{-1}$, hence for a large value of $\xi$ scattering between states that are far apart from each other in the in-plane momentum $k$ is exponentially suppressed $\sim \exp[-\xi^2(k-k')^2]$.

## 4.2 Scattering lengths

In the presence of surface and bulk states it is interesting to quantify averaged scattering rates and scattering probabilities between different types of states, which will be helpful to understand the numerical transport results.

To quantify the overall strength of impurity scattering, we define the averaged mean free path

$$l = \left\langle |v_\kappa| \left( \sum_{\kappa'} \delta(E_\kappa - E_F) q(\kappa, \kappa') \right)^{-1} \right\rangle, \tag{25}$$

where the Fermi-surface average is given by

$$\langle \ldots \rangle = \frac{1}{N} \sum_\kappa \delta(E_\kappa - E_F)(\ldots); \qquad N = \sum_\kappa \delta(E_\kappa - E_F), \tag{26}$$

and $N$ is the number of states at the Fermi level. Note that for a nearly constant $|v_\kappa|$ that we have, the mean free path is inversely proportional to the total scattering probability.

To quantify the scattering probability between different types of states $i \in [b, s]$, where $b$ ($s$) denotes bulk (surface) states, we define the scattering length

$$l_{ij} = \left\langle |v_\kappa| \left( \sum_{\kappa'}^j \delta(E_\kappa - E_F) q(\kappa, \kappa') \right)^{-1} \right\rangle_i, \tag{27}$$

where the sum runs over the type $j$ of states and the averaging is analogous to (26) but only over type $i$ of states ($\sum_\kappa \to \sum_\kappa^i$).

We now want to determine the dependence of the scattering probability on the width $W$ of the slab in two limiting cases (i) the number of bulk states being much larger than that of surface states, $N \gg N_s$ and (ii) the number of states being dominated by surface states, $N \sim N_s$. In regime (ii) the number of bulk states $N_b = N - N_s$, is given by the number of chiral bulk states. Thus the scaling of $N_b$ with the slab width reads

$$N_b \sim \begin{cases} W & \text{(i)} \\ 1 & \text{(ii).} \end{cases} \qquad (28)$$

The scaling of the scattering rate (24) is governed by the $z$ dependence of the wavefunctions. A normalized surface state with a penetration depth $\lambda \sim \beta^{-1} \sim 1$ and a normalized bulk state are of the form

$$\psi_s(z) = \sqrt{\frac{2}{\lambda}} \, e^{-z/\lambda}, \qquad\qquad \psi_b(z) = 1/\sqrt{W}, \qquad (29)$$

respectively. Consequently, for $W \gg \lambda$ (which we always consider), we can estimate the scaling of the scattering rate between the different types of states as

$$q(\boldsymbol{\kappa}, \boldsymbol{\kappa}') \sim \begin{cases} 1 & \text{same surface} \\ \frac{1}{W} & \text{bulk-surface, bulk-bulk} \\ 0 & \text{opposite surfaces.} \end{cases} \qquad (30)$$

From this, (28), and (27) the width dependence of the scattering probabilities summarizes to

|       | $l_{bb}/l$ | $l_{ss}/l$ | $l_{sb}/l$ | $l_{bs}/l$ |
|-------|------------|------------|------------|------------|
| (i)   | $1$ | $\frac{N_b}{N_s W} \sim 1$ | $1$ | $\frac{N_b}{N_s} \sim W$ |
| (ii)  | $W \frac{N_s}{N_b}$ | $1$ | $W \frac{N_s}{N_b}$ | $W$ |

$(31)$

Most importantly, in the regime (i) the scattering probability from bulk to surface ($\propto l_{bs}^{-1}$) is a factor $W$ smaller than other scattering probabilities, due to the ratio of the number of bulk states to surface states, which is large and increases with $W$. In the regime (ii) instead, the number of bulk states (consisting only of the chiral bulk states) does not depend on $W$. The peculiarity of this regime is that surface states scatter most probably within the surface states, which is due to the larger overlap of surface wavefunctions.

# 5 Numerical results

We calculate the nonequilibrium occupation function (13) at zero temperature by numerically solving Eq. (14) with respect to the transport length on the basis of the numerical solution of the discretized slab spectrum discussed in Section III [40]. The nonequilibrium occupation function (or, equivalently, the transport length) determines the conductivity, given in Eq. (17), and the valley polarization, defined in (5.2) below.

## 5.1 Conductivity

We consider the conductivity in units of the standard Drude estimate given by the mean free path $l$, the density of states at the Fermi level $n = N/\mathcal{V}$, and the Fermi velocity $v \approx t/\hbar$,

$$\sigma_0 = \frac{e^2 n l v}{3}. \qquad (32)$$

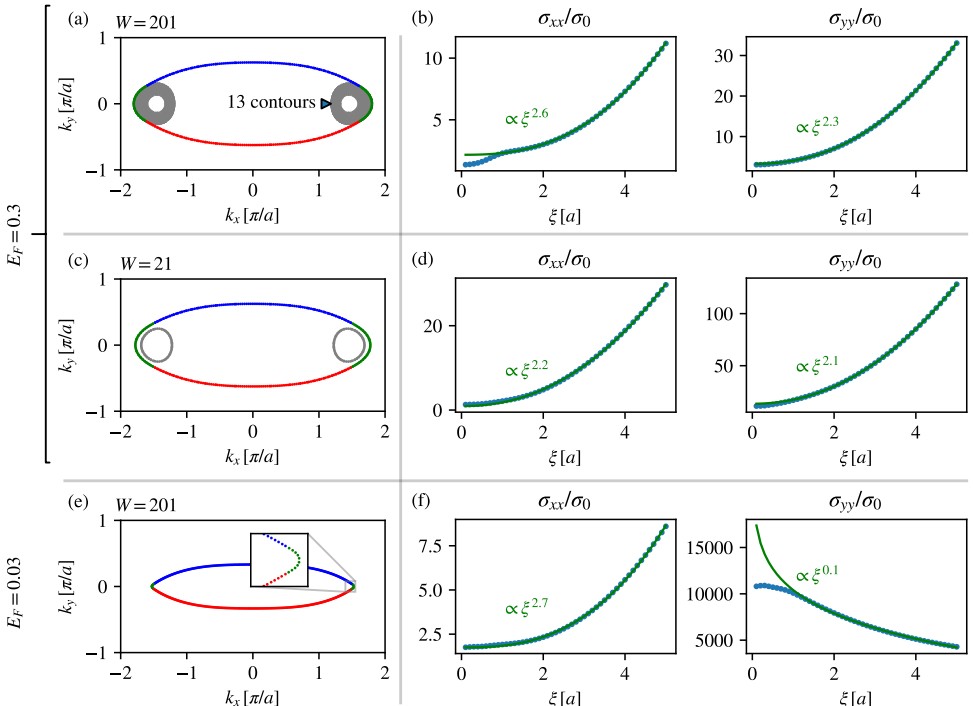

Figure 4: Impurity-range dependence of the conductivity. (a),(c),(e): Discretized Fermi-level contours for the three parameter choices ($E_F = 0.3\,t$, $W = 201$), ($E_F = 0.3\,t$, $W = 21$), and ($E_F = 0.03\,t$, $W = 201$). The number of finite-size induced states (surface states and chiral bulk states), $N_s + N_c$, relative to the total number of states $N$ is (a) $(N_s + N_c)/N = 0.14$, (c) $(N_s + N_c)/N = 0.68$, and (e) $(N_s + N_c)/N = 1$. Color indicates surface states at the upper (blue) and lower (red) surfaces, bulk states (gray), and the chiral bulk states (green), cf. Fig. 2. (b), (d), (f): Conductivity as a function of the impurity width $\xi$. The green curves show power-law fits, the exponent is indicated in the plot. Other parameters are $\beta = 1.5$, $\mu_b = 0.3\,t$.

This is the result one would expect to find for a system with point-like impurities ($\xi \to 0$) and only bulk states arranged in form of a spherical Fermi surface.

The dependence of the slab conductivity on the width of the impurity potential $\xi$ is summarized in Fig. 4. For $\xi > 1$ the conductivity is well fitted by a power-law dependence on $\xi$, with an exponent between 2 and 3. An exception is found for $\sigma_{yy}$ at $E_F \ll t$, which shows a weak $\xi$ dependence at a strongly enhanced conductivity at all $\xi$. In total, the magnitude of the conductivity, especially in the direction of motion of Fermi arcs, may be enhanced by several orders of magnitude, either due to a wide impurity range, or if the Fermi energy is close to the Weyl points.

To gain further insight, in Fig. 5 we consider the width dependence of the conductivity. Figure 5(a) shows that in the case of a large number of bulk states and point-like impurities, the conductivity is nearly independent of $W$ and is close to $\sigma_0$ — in this regime the slab thus resembles a conventional metal. At a large $\xi$, however, the conductivity enhancement decreases antiproportional to the width, indicating that the conductivity enhancement at large $E_F$ is related to the presence of the confinement-induced surface and chiral bulk states, which number, relative to the total number of states $N$, is antiproportional to $W$. At small $E_F$, however, when there is only one Fermi-level contour mainly consisting of surface states, the total number of states is nearly independent of $W$. In this case, the large conductivity in the direction of motion of surface states, $\sigma_{yy}$, linearly *increases* with the width $W$.

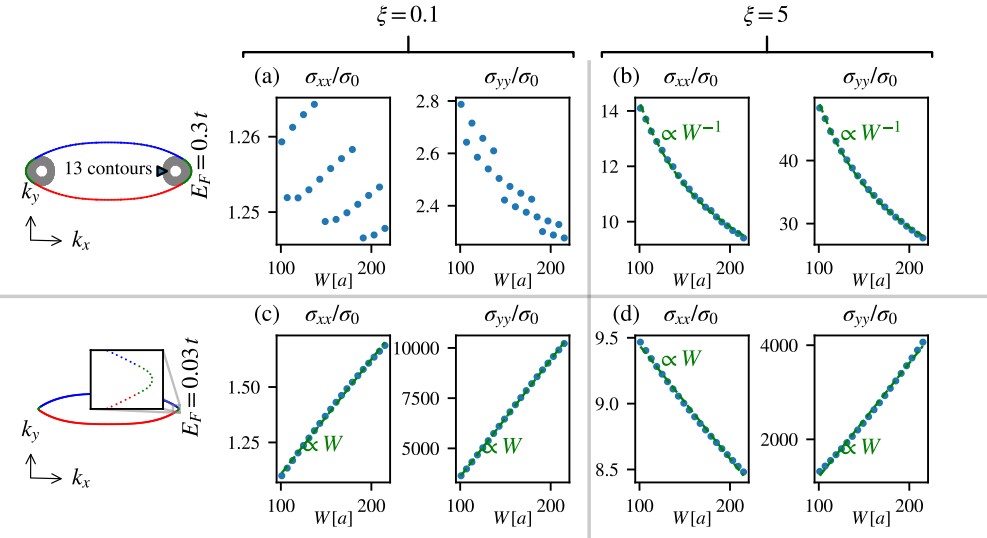

Figure 5: Width dependence of the conductivity at (a) $E_F = 0.3\,t$, $\xi = 0.1$, (b) $E_F = 0.3\,t$, $\xi = 5$, (c) $E_F = 0.03\,t$, $\xi = 0.1$, and $E_F = 0.03\,t$, $\xi = 5$. Other parameters are $\beta = 1.5$, $\mu_b = 0.3\,t$. The dashed lines in (b) indicate $W^{-1}$ dependence and in (c) and (d) linear dependence.

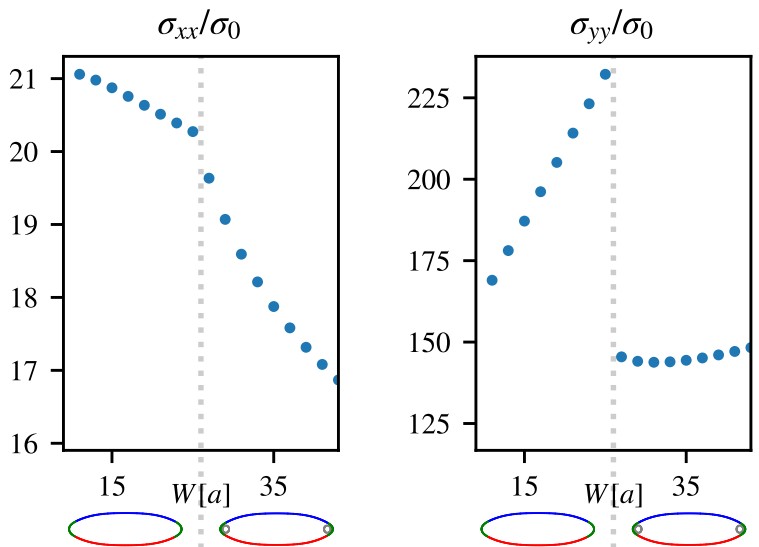

Figure 6: Width dependence of the conductivity at $\xi = 5$ and $E_F = 0.15\,t$, $\mu_b = 0.3\,t$. The transition from a single contour to three contours is indicated by the dotted lines and the contour-plots below the $W$ axis.

To explore more carefully the transition from an increasing to a decreasing $W$ dependence, in Fig. 6 we plot the conductivity at smaller width, when the first bulk contours appear. This plot shows that the strong $W$ enhancement of the surface conductivity requires the normal bulk contours to vanish, which happens if $W k_F / \pi \lesssim 1$. As soon as at least one normal bulk contour appears, the conductivity jumps to a significantly lower value and becomes decreasing in $W$.

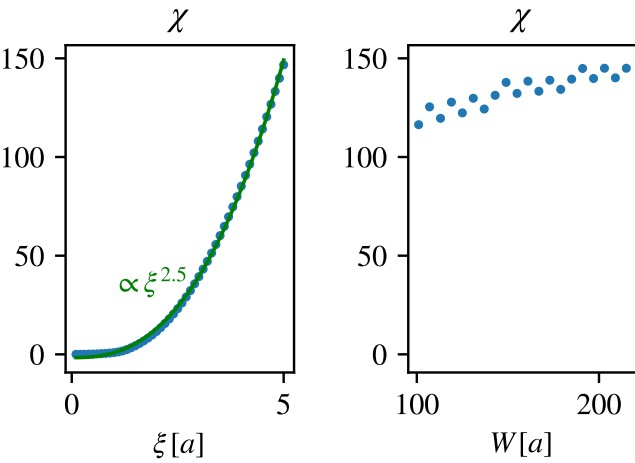

Figure 7: Valley polarization $\chi$, defined in Eq. (5.2). (a) Dependence of $\chi$ on the impurity range $\xi$ at a width $W = 201$. (b) Width dependence of $\chi$ at $\xi = 5$. Other parameters are $\mu_b = 0.3\,t$, $\beta = 1.5$, and $E_F = 0.3\,t$.

## 5.2 Valley polarization

Besides the conductivity it is interesting to explore the non-equilibrium occupation difference of the valleys, which occurs when the electric field points along the valley separation, $\boldsymbol{E} = \hat{\boldsymbol{x}}E$. The average occupation of the valley at $k_x = \pm\beta$ is given by $\sum_{\boldsymbol{\kappa}}^{\pm} g_{\boldsymbol{\kappa}}/(N_b/2)$, where the sum runs over all bulk states at the valley $\pm$, and $N_b/2 = \sum_{\boldsymbol{\kappa}}^{\pm}$ is the number of those states. We quantify the valley polarization by the difference of the valley occupations relative to the standard occupation difference of states $\delta(E_F - E_{\boldsymbol{\kappa}})eEl$ due to the mean free motion in the electric field,

$$\chi \equiv \frac{\sum_{\boldsymbol{\kappa}}^{+} g_{\boldsymbol{\kappa}}}{\sum_{\boldsymbol{\kappa}}^{+} \delta(E_F - E_{\boldsymbol{\kappa}})eEl} - \frac{\sum_{\boldsymbol{\kappa}}^{-} g_{\boldsymbol{\kappa}}}{\sum_{\boldsymbol{\kappa}}^{-} \delta(E_F - E_{\boldsymbol{\kappa}})eEl} = \frac{\langle \Lambda_{\boldsymbol{\kappa}}^x \rangle_+ - \langle \Lambda_{\boldsymbol{\kappa}}^x \rangle_-}{l}, \tag{33}$$

where in the second line we used Eq. (13) and defined the average over valley bulk states $\langle\dots\rangle_{\pm} = \sum_{\boldsymbol{\kappa}}^{\pm} \delta(E_F - E_{\boldsymbol{\kappa}})\dots/\sum_{\boldsymbol{\kappa}}^{\pm} \delta(E_F - E_{\boldsymbol{\kappa}})$.

A representative result for the impurity range and width dependence of the valley polarization is shown in Fig. 7. The valley polarization shows the power-law $\xi$ dependence, similar to the conductivity. However, unlike for the conductivity, there is no significant width dependence at large $\xi$ (here already at $\xi \gtrsim 1$) in the numerically accessible width range. This is very surprising as it seems to imply a presence of valley polarization for sufficiently large impurity widths in arbitrary large systems, contradicting previous predictions based on infinite-system calculation [3]. Below we will show that the valley polarization in fact does decay but for width above $\gtrsim l\,e^{(\xi\Delta k)^2}$, which becomes exponentially large for $\xi\Delta k \gtrsim 1$.

## 6 Discussion

The numerical results of the previous section show enhancements of the slab conductivity by several orders of magnitude (compared to the expectation for a conventional metal (32)) and a substantial valley polarization in a wide region of the parameter space. The characteristic dependencies on $\xi$ and $W$ allow to identify the main mechanisms of these effects, which we now systematically discuss.

## 6.1 Impurity-range dependence

When the impurity range $\xi$ increases, the scattering rate (24) between two countermovers separated by $\Delta k$ becomes exponentially suppressed, $\sim \exp[-(\xi\Delta k)^2]$. The transport length does not inherit the exponential enhancement though, since relaxation happens via multiple small-angle scattering processes. To illustrate this, we consider a toy model of a closed chain of $\mathcal{N}$ states labeled $i \in [1, \mathcal{N}]$ with arbitrary velocities $\boldsymbol{v}_i$. The BE (14) is of the form (summation over repeated indices assumed)

$$\boldsymbol{v}_i = M_{ij}\boldsymbol{\Lambda}_j, \tag{34}$$

where $M_{ij} = \delta_{ij}\sum_k q_{ik} - q_{ij}$ is given by the scattering rates between states, $q_{ij}$. We assume scattering only between the nearest neighbors, with the rate $q_{\mathrm{nn}}$, and direct scattering between countermovers with the rate $q_{\mathrm{d}}$, in which case the matrix $M$ becomes

$$M_{ij} = (2q_{\mathrm{nn}} + 2q_{\mathrm{d}})\delta_{ij} - q_{\mathrm{nn}}(\delta_{ij+1} + \delta_{i,j-1}), \tag{35}$$

where we used that the direct-scattering part of $q_{ij}$ cancels when multiplied with $\boldsymbol{\Lambda}$. For the nearest-neighbor part $M_{\mathrm{nn}}$ of $M$ there is a left pseudo-inverse,

$$P_{ij} = \frac{1}{q_{\mathrm{nn}}}\frac{|i-j|(|i-j|-\mathcal{N})}{2\mathcal{N}}, \tag{36}$$

so that $PM_{\mathrm{nn}}\boldsymbol{\Lambda} = \boldsymbol{\Lambda}$ (note $\sum_i \Lambda_i = 0$ due to particle conservation). With its help, the full solution of (34) becomes

$$\boldsymbol{\Lambda} = (1 + 2q_{\mathrm{d}}P)^{-1}P\boldsymbol{v}. \tag{37}$$

We first ignore direct scattering. From the form of $P$, it is clear that the solution can depend on $\mathcal{N}$ to maximally the power $\mathcal{N}^2$. In particular, for a circular velocity arrangement $\boldsymbol{v} = \{-\cos[2\pi(i-1)/\mathcal{N}], \sin[2\pi(i-1)/\mathcal{N}]\}$ the vector mean free path for $\mathcal{N} \gg 1$ assumes the value

$$\boldsymbol{\Lambda} = \frac{1}{q_{\mathrm{nn}}}\frac{(\tilde{\xi}\Delta k)^2}{4}\boldsymbol{v}, \tag{38}$$

where we have written the number of nearest neighbors as $\mathcal{N} = \pi\tilde{\xi}\Delta k$, in terms of the spacing between nearest neighbors $\tilde{\xi}^{-1}$ and the distance between countermovers $\Delta k$.

Considering the full result with direct scattering in Eq. (37), we see that the nearest-neighbor scattering contributions to $\boldsymbol{\Lambda}$ dominate as long as

$$\frac{(\tilde{\xi}\Delta k)^p}{q_{\mathrm{nn}}} \ll \frac{1}{q_{\mathrm{d}}}, \tag{39}$$

where the exponent $p$ depends on the velocity arrangement. In the opposite limit we instead obtain

$$\boldsymbol{\Lambda} = \frac{1}{2q_{\mathrm{d}}}\boldsymbol{v}. \tag{40}$$

Transferring this insight to the Weyl slab model, the distance between the nearest neighbors $\tilde{\xi}^{-1}$ corresponds to $\xi^{-1}$. The mean free path corresponds to the inverse diagonal of $M$ times velocity, $l \sim v/(q_{\mathrm{d}} + q_{\mathrm{nn}})$. Interpolating between the two regimes of dominant direct scattering and nearest-neighbor scattering the typical values of the transport length may be well estimated as

$$\bar{\Lambda} = l\left[1 + (\xi\Delta k)^p\right]. \tag{41}$$

This explains the general power-law conductivity enhancement with $\xi$ in Fig. 4, except for $\sigma_{yy}$ in Fig. 4(f), which we discuss separately.

## 6.2 Valley polarization

The averaged occupation of the two valleys $\pm$ can be expressed as

$$\mu_\pm = \frac{\sum_{\kappa}^{\pm} g_\kappa}{N_b/2} , \tag{42}$$

where $\sum_{\kappa}^{\pm} \delta(E_F - E_\kappa) = N_b/2$. The occupation difference is related to the valley polarization $\chi$ defined in Eq. (5.2),

$$\mu_+ - \mu_- = \chi \, e E_x l . \tag{43}$$

A single valley has a total velocity in the $x$ direction — the unbalanced velocity of the chiral states $v N_c/2$, where $N_c$ is the total number of chiral states ($N_c/2$ in each valley). An electric field in the $x$ direction thus pumps charge between the valleys with the rate $v e E_x N_c$, which in a steady state must be counterbalanced by scattering. We can write down a simple balance equation as a condition for a steady valley occupation,

$$e N_b \frac{d\mu_\pm}{dt} = 0 = \pm v e E_x N_c + e N_b \left( \frac{\partial \mu_\pm}{\partial t} \right)_{\text{scat}} , \tag{44}$$

where we used $N_b \gg N_c$. The time-change of $\mu_+ - \mu_-$ due to scattering is proportional to the occupation difference itself and the scattering probability, which we quantify by the scattering length $l_c$, hence

$$\left( \frac{\partial (\mu_+ - \mu_-)}{\partial t} \right)_{\text{scat}} = -2 \frac{\mu_+ - \mu_-}{l_c/v} = -2 \frac{\chi e E l}{l_c/v} . \tag{45}$$

Together with (44) we obtain

$$\chi = \frac{N_c}{N_b} \frac{l_c}{l} . \tag{46}$$

For a point-like disorder potential ($\xi \to 0$) the ratio $l_c/l$ goes to one and the valley polarization is small. For larger $\xi$, direct scattering between the valleys becomes strongly suppressed and the relaxation of $\mu_+ - \mu_-$ must go via surface states. Thereby the relaxation along the arcs and the scattering from surface to bulk is much faster than the scattering from bulk to surface, see (31). The scattering length $l_c$ is thus set by the bulk-surface scattering length $l_{bs}$, which is proportional to the ratio $N_b/N_s$ so that

$$\chi \propto \frac{N_c}{N_s} . \tag{47}$$

This explains the surprising result that the valley polarization does not depend on the width, as seen numerically in Fig. 6. This is surprising since the origin of the valley polarization are the chiral bulk states, which number $N_c$ is a factor $\propto W$ smaller than the total number of bulk states $N_b$. The explanation is that the valley relaxation also becomes suppressed $\propto W^{-1}$ since the probability to scatter into Fermi arcs decreases with an increasing number of bulk states. For a much larger width, when the probability of relaxation via Fermi arcs becomes smaller than the probability of relaxation via direct inter-valley scattering, the valley polarization will ultimately go to zero like $W^{-1}$. In case of a Gaussian potential, the amplitude of direct inter-valley cattering is however exponentially suppressed $\sim \exp[-(\xi \Delta k)^2]$ so that the width independence can easily extend to arbitrary macroscopic sizes for realistic values of the impurity width and the cone separation $\xi \Delta k \gtrsim 1$.

Regarding the strong enhancement of valley polarization with $\xi$, one is tempted to understand it as a consequence of an increasing number of scattering events needed for a relaxation along the Fermi arcs. However, the contribution of such a process to $\chi$ would enter in the form $l_{ss}(\xi \Delta)^p/l \propto 1/W$, with a clear $1/W$ dependence, which we do not observe. It rather must be

the suppression of the bulk-surface scattering probability with $\xi$, which is plausible in view of the typical effect of an increasing $\xi$ to increase the relaxation time. We note, however, that the simplified analytical calculation of Section 6.1 does not apply since in this case the valleys correspond to two-dimensional pools of states, while the model of Section 6.1 only considers one-dimensional chains.

### 6.3 Conductivity in case of a large number of bulk states

For bulk states (excluding chiral bulk states) the small separation of countermovers in momentum space $\sim k_F$ makes their transport length close to the mean free path $l$ and not significantly enhanced with $\xi$. We thus approximate the current contribution of bulk states as

$$\boldsymbol{j}_n = \sigma_0 \boldsymbol{E}. \tag{48}$$

For an electric field in the $y$ direction there is additionally the contribution of surface states, which transport length is mainly set by surface-surface scattering, enhanced by $\xi$ according to (41),

$$\boldsymbol{j}_s = e^2 n_s v \, l_{ss} (\xi \Delta k)^p \boldsymbol{y} E_y \,, \tag{49}$$

where $n_s = N_s/V$ is the density of surface states.

The current contribution of chiral bulk states is negligible compared to (48), except if the valley polarization becomes large, in which case

$$\boldsymbol{j}_c = e v \hat{\boldsymbol{x}} n_c (\mu_+ - \mu_-)/2 = e^2 v \hat{\boldsymbol{x}} n_c \chi l \, E/2 \,, \tag{50}$$

where $n_c = N_c/V$ is the density of chiral bulk states. Adding all the current contributions, we obtain

$$\frac{\sigma_{xx}^{(i)}}{\sigma_0} = 1 + \frac{3}{2} \frac{N_c}{N_b} \chi \,, \tag{51}$$

$$\frac{\sigma_{yy}^{(i)}}{\sigma_0} = 1 + 3 \frac{N_s}{N_b} \frac{l_{ss} (\xi \Delta k)^p}{l} \sim 1 + \frac{1}{W} (\xi \Delta k)^p \,, \tag{52}$$

where we again used (31). This rough estimate is in qualitative agreement with the numerical results. Note that $N_b$ increases proportional to $W$, which explains the $W$ dependence of $\sigma_{xx}$ and $\sigma_{yy}$ in Fig. 5(b). We now understand that the enhancement of $\sigma_{xx}$ and $\sigma_{yy}$ are mainly due to chiral bulk states and surface states, respectively.

In conventional metals the conductivity is width independent as long as the width is much larger than the mean free path; when the width becomes smaller, the conductivity tends to decrease due to additional scattering at boundaries. Our work shows that in Weyl semimetals away from charge neutrality the opposite trend of increasing conductivity with shrinking the width may occur due to chiral bulk states and surface states. A qualitatively similar width dependence may occur also in the regime $l \ll W$ (we consider the opposite limit $l \gg W$), which has been considered in Ref. [34]. Observations of enhanced conductivity for reduced widths have been reported in Ref. [35], where the Weyl semimetal nanobelts, according to estimates of the mean free path, are presumably in the regime of our work, $l \gg W$, or in the crossover regime $l \sim W$.

### 6.4 Conductivity in case of a small number of bulk states

We now come to the case (ii) when the number of states is dominated by surface states, while in the bulk only chiral bulk states are present. The conductivity in the $y$ direction can be written in the form

$$\frac{\sigma_{yy}^{(ii)}}{\sigma_0} = 1 + \frac{N_s}{N} \frac{\Lambda_s}{l} \,, \tag{53}$$

where $N_s \approx N$ is the number and $\Lambda_s$ the transport length of surface states. Scattering within surface states at the same surface does not lead to relaxation of motion in the $y$ direction since the average velocity $v_y$ of those states is not zero. Countermoving surface states, on the other hand, have no overlap with each other, direct scattering between them is blocked. The relaxation of Fermi arc states must thus go via the small number of chiral bulk states, so that $\Lambda_s$ is set by the surface-bulk scattering probability, $\Lambda_s \sim l_{sb}$. In Section 4.2 we found $l_{sb}/l \sim WN_s/N_c$—Eq. (31)—which leads to

$$\frac{\sigma_{yy}^{(ii)}}{\sigma_0} \sim W \frac{N_s}{N_c} \ . \tag{54}$$

Both factors are of order 100 for parameters in Fig. 4(f), which explains the large magnitude. Also the $W$ dependence in Fig. 5 is consistent since both $N_s$ and $N_c$ are $W$ independent in this case.

For large $\xi$, surface-bulk scattering becomes limited to small regions at the nodes and the full relaxation must involve nearest-neighbor scattering along the Fermi arc. According to Section 6.1, the latter should elongate the full transport length by an additional $l_{ss}(\xi\Delta k)^p$. For the considered parameters, $l_{sb}$ is much larger than this additional part since $(\xi\Delta k)^p \ll WN_s/N_c$, which explains the weak $\xi$ dependence in Fig. 4(f).

The transition from (ii) to (i) spoils the strong enhancement of $\sigma_{yy}$ in two ways: First, the ratio $N_s/N_c$ changes to $N_s/(N_c + N_b)$ and thus becomes smaller and second, according to Eq. (31), $l_{sb}/l \sim WN_s/N_b \to 1$ is no longer width dependent, which in Fig. 6 explains the jump and the change of slope.

The conductivity in the $x$ direction is also governed by the dominant number of surface states. Relaxation however happens via scattering within the same surface, since $v_x$ averages to zero at each surface separately. Since $l_{ss}/l \sim 1$, the conductivity in the $x$ direction is not significantly enhanced.

# 7  Conclusion

In conclusion, we have studied linear-response properties of a finite Weyl semimetal slab (width $W$) in the presence of long-ranged disorder (disorder potential width $\xi$). Our work highlights the remarkable property of Weyl semimetals to realize valleys of opposite chirality that are well separated in momentum space ($\Delta k$) and continuously connected only via surface states. For a Fermi energy that is not exactly at the Weyl nodes, the surface states occur together with confinement-induced chiral bulk states. In the presence of an electric field parallel to cone separation they allow to violate chiral charge conservation even without an external magnetic field. This peculiarity stabilizes an anomalous valley polarization at zero magnetic field. If the potential width is substantially larger than the inverse separation of valleys, the valley polarization persists up to very large slab width. This is explained by the fact that direct inter-valley scattering is strongly suppressed and the relaxation must go via Fermi arcs, which is however also increasingly ineffective owing to their vanishing density with an increasing width. The resulting width independence of the confintenment-induced valley polarization persists up to a width, for which relaxation via direct inter-valley scattering becomes more effective than relaxation via Fermi arcs. For Gaussian-type disorder potentials this maximum width is exponentially enhanced by $\exp[(\xi\Delta k)^2]$ and can thus easily reach macroscopic length scales at realistic values of cone separation $\Delta k$ and inpurity-potential widths $\xi$.

The valley polarization and Fermi-arc surface states lead to a conductivity enhancement which increases with an increasing width of the disorder potential and a decreasing width of the slab ($\sigma \propto 1/W$). Moreover, if the Fermi energy is reduced towards charge neutrality such

that normal bulk states vanish completely, the conductivity in the direction of motion of surface states becomes strongly enhanced because relaxation of surface states can only go via bulk states which number becomes strongly reduced.

Methodologically our work performs first steps in the application of the weak-disorder transport formalism to a multilayer system with a large ($\gtrsim 100$) number of layers and consequently a similarly large number of bands in the in-plane Brillouin zone. The numerical code [40] is designed to be easily applicable to an arbitrary lattice model and can thus be used to explore in detail the confinement-induced valley polarization in various Weyl-metal models. In this work, we introduced the formalism by considering a minimal two-Weyl-cone model. We find that the qualitative aspects of the valley polarization are robust to lattice details such as boundary potentials (which give the Fermi arcs a finite curvature) or velocity anisotropy of the Weyl cones. Our analytical discussion shows that the valley polarization depends on the mere presence of chiral bulk states and surface states (which is topological) and the ratio of the inverse separation of Weyl cones vs. the width of the scattering potential, which explains the robustness of this effect.

An interesting application of the introduced tools is to consider lattice models of existing Weyl semimetals. For the case of several pairs of Weyl nodes that are sufficiently separated in momentum space, such as in the TaAs material family, we expect valley polarization to occur in each pair which cone sepration aligns with the electric field, similarly to the two-cone case. The reason is that due to the large pair separation, scattering between pairs should be negligible compared to the intervalley scattering within a single pair, making each pair independent and thus reduce the problem to the two-cone case.

General limitations of the introduced numerical tools are the restriction to slab width being smaller than the mean free path and the restriction to the leading order in the disorder potential. Both the fate of confinement-induced effects for larger widths as well as corrections of higher order in the disorder potential, which are known to start with Berry phase effects [44], constitute interesting directions to extend this formalism.

### Data availability

All the code and data used to produce the reported results is available in Ref. [40].

### Author contributions

M.B. formulated the project idea, developed the theory with input from N.B. and A.A., performed and analyzed numerical calculations, and developed the analytical model. P.M.P.P. developed the numerical code with input from N.B., A.A., and M.B., and performed and analyzed numerical calculations. The manuscript was written by M.B. with input from P.M.P.P. and A.A.

# Acknowledgments

This research was supported by the European Union Horizon 2020 research and innovation programme under Grant Agreement No. 824140, Grant No. 18688556 of the Deutsche Forschungsgemeinschaft (DFG, German Research Foundation), ERC Starging Grant 638760, and the Netherlands Organisation for Scientific Research (NWO/OCW), as part of the Frontiers of Nanoscience program.

## Supplementary Material

### Derivation of scattering amplitudes

We consider Gaussian-type static impurity potentials,

$$V = \sum_\alpha U_\alpha \phi(\mathbf{r} - \mathbf{r}_\alpha), \qquad\qquad \phi(\mathbf{r}) = e^{-|\mathbf{r}|^2/2\xi^2}, \qquad\qquad \text{(S1)}$$

where the sum runs over impurities with a characteristic width $\xi$, random and uncorrelated potential magnitudes $U_\alpha \in [-\delta, \delta]$, and random positions $\mathbf{r}_\alpha$.

For our transport consideration we consider the scattering rate $Q(\boldsymbol{\kappa}, \boldsymbol{\kappa}') = 2\pi\delta(E_{\boldsymbol{\kappa}} - E_F) \times q(\boldsymbol{\kappa}, \boldsymbol{\kappa}')$ between energy eigenstates $|\psi_{\boldsymbol{\kappa}}\rangle$ and $|\psi_{\boldsymbol{\kappa}'}\rangle$, which we calculate using Fermi's Golden Rule,

$$q(\boldsymbol{\kappa}, \boldsymbol{\kappa}') = \left\langle\!\left\langle |\langle\psi_{\boldsymbol{\kappa}}|V|\psi_{\boldsymbol{\kappa}'}\rangle|^2 \right\rangle\!\right\rangle, \qquad\qquad \text{(S2)}$$

where the disorder average is defined as

$$\left\langle\!\left\langle \ldots \right\rangle\!\right\rangle = \prod_\alpha \int_{-\delta}^{\delta} \frac{dU_\alpha}{2\delta} \int \frac{d\mathbf{r}_\alpha}{\mathcal{V}} (\ldots). \qquad\qquad \text{(S3)}$$

We write the normalized wavefunctions as

$$\langle\mathbf{r}|\psi_{\boldsymbol{\kappa}}\rangle = \frac{1}{\sqrt{L_x L_y}} e^{i\mathbf{k}\cdot\boldsymbol{\rho}} \, \psi_{\boldsymbol{\kappa}}(z), \qquad\qquad \text{(S4)}$$

where $\boldsymbol{\rho} = (x, y)$ is the in-plane position and $L_x L_y$ the in-plane volume and $\psi_{\boldsymbol{\kappa}}(z)$ is the normalized eigenvector of numerical diagonalization of the lattice model, $z$ denoting the discrete sites in the $z$ direction.

The expectation value of the impurity then calculates to

$$\langle\psi_{\boldsymbol{\kappa}'}|V|\psi_{\boldsymbol{\kappa}}\rangle = \frac{2\pi\xi^2}{L_x L_y} e^{-\xi^2(\mathbf{k}-\mathbf{k}')^2/2} \sum_\alpha U_\alpha \sum_z \phi(z - z_\alpha) \psi_{\boldsymbol{\kappa}'}^\dagger(z)\psi_{\boldsymbol{\kappa}}(z). \qquad \text{(S5)}$$

Inserting into (S2) and using (S3), we obtain

$$
\begin{aligned}
q(\boldsymbol{\kappa}, \boldsymbol{\kappa}') &= \frac{4\pi^2\xi^4}{L_x^2 L_y^2} e^{-\xi^2(\mathbf{k}-\mathbf{k}')^2} \left\langle\!\!\left\langle \left(\sum_{\alpha,\beta} U_\alpha U_\beta \sum_{z,z'} \int \frac{dz_\alpha}{W} \phi(z-z_\alpha)\phi(z'-z_\beta)\psi_{\boldsymbol{\kappa}'}^\dagger(z)\psi_{\boldsymbol{\kappa}}(z)\psi_{\boldsymbol{\kappa}}^\dagger(z')\psi_{\boldsymbol{\kappa}'}(z')\right)\right\rangle\!\!\right\rangle \\
&= \frac{4\pi^2\xi^4\delta^2}{3L_x^2 L_y^2} e^{-\xi^2(\mathbf{k}-\mathbf{k}')^2} \sum_\alpha \sum_{z,z'} \psi_{\boldsymbol{\kappa}'}^\dagger(z)\psi_{\boldsymbol{\kappa}}(z)\psi_{\boldsymbol{\kappa}}^\dagger(z')\psi_{\boldsymbol{\kappa}'}(z') \int \frac{dz_\alpha}{W} \phi(z-z_\alpha)\phi(z'-z_\alpha) \\
&= \frac{4\pi^2\xi^4\delta^2 n_i}{3L_x L_y} e^{-\xi^2(\mathbf{k}-\mathbf{k}')^2} \sum_{z,z'} \psi_{\boldsymbol{\kappa}'}^\dagger(z)\psi_{\boldsymbol{\kappa}}(z)\psi_{\boldsymbol{\kappa}}^\dagger(z')\psi_{\boldsymbol{\kappa}'}(z') \int dz_i \, \phi(z-z_i)\phi(z'-z_i) \\
&= \frac{(2\pi\xi^2)^3\delta^2 n_i}{3L_x L_y} e^{-\xi^2(\mathbf{k}-\mathbf{k}')^2} \sum_{z,z'} \psi_{\boldsymbol{\kappa}'}^\dagger(z)\psi_{\boldsymbol{\kappa}}(z)\psi_{\boldsymbol{\kappa}}^\dagger(z')\psi_{\boldsymbol{\kappa}'}(z') M(z, z', \xi, W),
\end{aligned}
$$
$$\text{(S6)}$$

where $n_i = \sum_\alpha / L_x L_y W$ is the impurity concentration, furthermore we used

$$\left\langle\!\left\langle U_\alpha U_\beta \ldots \right\rangle\!\right\rangle = \delta_{\alpha\beta} \frac{\delta^2}{3} \left\langle\!\left\langle \ldots \right\rangle\!\right\rangle, \qquad\qquad \text{(S7)}$$

and defined the function

$$M(z, z', \xi, W) = \frac{1}{2\pi\xi^2} \int dz_i \, \phi(z-z_i)\phi(z'-z_i) \qquad\qquad \text{(S8)}$$

$$= \frac{e^{-(z-z')^2/4\xi^2}}{4\sqrt{\pi}\xi} \left[ \text{erf}\left(\frac{W+z+z'}{2\xi}\right) + \text{erf}\left(\frac{W-z-z'}{2\xi}\right) \right], \qquad \text{(S9)}$$

plotted in Fig. 3; the error function is defined as $\mathrm{erf}(x) = (2/\sqrt{\pi}) \int_0^x e^{-t^2} dt$. Note that

$$\int dz' \, M(z, z', \xi, W) \tag{S10}$$

is a weak function of $z$, equal to 1 in the middle of the slab, and going down to $\approx 0.4$ at the edges in the range $\xi$.

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
