# Peer review of "Chiral Anomaly Trapped in Weyl Metals: Nonequilibrium Valley Polarization at Zero Magnetic Field"

_SciPost Physics, doi:SciPost Phys. 11, 046 (2021)_

## Round 1 · Referee Report · Titus Neupert (Referee 1) · 2021-5-20

Strengths

1) clear writing and good presentation 2) methodological advancement combined with nice physics question 3) many future applications and extensions for the method are conceivable

Report

The manuscript introduces a formalism for the numerical computation of transport properties in mesoscopic systems for weak potential scattering. This formalism is applied to thin slabs of a Weyl semimetal, demonstrating a conductivity enhancement not dissimilar to the chiral anomaly, but purely introduced by finite size quantization. The manuscript is very well written and organized. It addresses a relevant physics question while at the same time introducing novel methodology.

I have three clarification questions and a few concrete requests for changes. The questions are: 1) Is the algorithm that finds the states on the Fermi contours guaranteed to produce the correct density of states for each band, i.e., does it take into account the magnitude of the fermi velocity when producing the discretization? 2) Much of the arguments depend on the numbers N, Ns, Nb, W. For the simulations, we only learn about the W that has been used. Maybe the Ns, Nb, N (or their ratio) could also be incorporated in the figures (maybe visually) if the authors would also consider that beneficial for the reader. 3) The method as presented in Sec. IIA would also be applicable to 3D systems, upon introducing another momentum quantum number. I do not understand why, as stated in Sec. IIB, it becomes invalid for thick slabs. In this case, should there not be an 'emergent' momentum quantum number, the kz momentum, so that the approach remains valid? Asked the other way around: if one "forgets" to include a quantum number in the formulation, would the numerics not work? 4) I find it a bit unsettling that the contradiction with the literature stated at the end of Sec. VB is left unresolved.

The requests for changes are noted below.

In view of the high quality of the manuscript, I would recommend its publication, provided the questions and requested changes are thoroughly addressed.

Requested changes

1) I think the work would benefit from a discussion on the robustness of the observed phenomena in more generic Weyl fermiologies: What if additional pockets are present in the bulk or on the surface, what if the Fermi arcs are not so perfectly straight (like in TaAs and similar materials, see e.g., PRB 97, 085142), what if the bulk Weyl cones are strongly anisotropic (while still type I) as is often the case?

2) For a paper that is on the technical side, I find the introduction and review of previous results too succinct. A more in-depth summary of results from the literature (which may also require extending the rather short list of references) would be beneficial for the reader.

3) There is a typo in the exponent of Eq. 29: x-> z

  • validity: high
  • significance: high
  • originality: high
  • clarity: high
  • formatting: excellent
  • grammar: excellent

Author:  Maxim Breitkreiz  on 2021-07-14  [id 1564]

(in reply to Report 1 by Titus Neupert on 2021-05-20)
Category:
answer to question

Thank you very much for the positive feedback and the useful suggestions.

In response to the comments in the report:

  1. Yes, the algorithm does take into account the true Fermi velocity. Only in the analytical treatment the velocity is assumed to be constant for simplicity.

  2. The precise values for the numbers of states are of course easily calculable from the solved slab model — it essentially corresponds to the lengths of the corresponding contours. In the new version we now provide numerical values in the caption of Fig. 4 to show concrete numbers and make contact with the discussion section. The arguments in the discussion section, however, only use rough and obvious properties of these numbers, such as the ratio N_s/N to be small for large width or the 1/W dependence of N_s/N, which does not seem to require extra plots.

  3. The slab kinetic equation is different compared to the infinite-system one and it is valid for thin slabs as long as the width is smaller than the mean free path, as shown in Section IIB On the other hand, it is indeed correct that the used kinetic equation is valid for an arbitrary number of infinite dimensions. One can thus expect that an infinite-system equation will give correct results for a thick slab, where the width is basically much larger than all other relevant length scales. It is however the purpose of our work to explore the effect of a finite size, specifically one of the three dimensions is considered finite (slab). The equation for a slab is different in that it accounts for confinement-induced states, which, as we show, give unique effects in transport. The question arises up to which slab width one should expect the slab behavior? This is answered in section IIB, where we show that the kinetic equation for a slab fails if the slab width W exceeds the scattering mean free path l. One can possibly save the equation by a transformation to the ``emergent quantum number’’ (the momentum in the out-of-plane direction) for which the position matrix element will become small instead of \sim W (which causes the failure of the slab equation). Will this transformation just lead to the infinite-system equation or can some confinement-induced effects survive? This is a very interesting question, which we would be eager to explore in future work.

  4. The contradiction is resolved by the fact that there exists a large length scale \sim l exp[-(\xi\Delta k)^2], which the width must exceed for the valley polarization to become suppressed. This is discussed not at the end of section V but in the discussion, section VI, below Eq. (47). In the new version we added a sentence at the end of section V to clarify that the seeming contradiction will be resolved below.

In response to requested changes:

  1. We clarified the robustness of the effect seen in the minimal model that we used and included a discussion of our expectations for other Weyl models in the conclusion section (third and second to last paragraphs). Moreover, in our model, the Fermi arcs are not straight: Our model includes variable boundary potentials, which curve the Fermi arcs, as explained in section III and seen e.g. in the contour plot of Fig. 2. The velocity is also not perfectly isotropic.

  2. We have thoroughly rewritten the introduction, in particular extending the review of the role of the valley degree of freedom, disorder, and finite-size effects. Thereby we added 14 new references [5, 7, 18-25, 28, 37-39].

  3. We corrected the typo, thank you!

---

## Round 1 · Referee Report · Anonymous (Referee 2) · 2021-6-28

Report

This paper is well-written and, taking into account the comments of Referee 1, the results in the paper are scientifically sound. I have a more general question: A random disorder potential should allow for bound states of Weyl fermions the valleys of the potential. A similar situation shows up in QCD, where quarks have a small mass and are approximately chiral Weyl fermions. Due to the confining potential for quarks generated by the strong force, left-handed and hence left-moving quarks are reflected into right moving and hence right-handed once at the right potential wall, go back to the left potential wall, get reflected back and so on. The potential walls hence mix left and right handed quarks, and the reflection back and forth leads to a non vanishing chiral condensate <PsibarL PsiR>. In condensed matter terms, this is a inter-valley pairing condensate. In QCD, this condensate spontaneously breaks the chiral symmetry (axial U(1) in Weyl semimetals). This argument is due to Banks and Casher. The upshot is that the bound states in a confining potential lead to a chiral condensate and hence to spontaneous symmetry breaking. The resulting goldstone modes are the mesons of QCD, and are approximately massless (due to the small quark mass). My question now is: If we have Weyl fermions trapped in a random disorder potential, a similar inter valley pairing condensate should be formed, in particular in the limit that the authors consider (Delta k >> 1/xi). This condensate should contribute to the conductivity calculation. Is this taken into account in this work already? If this question is satisfactorily answered, I support publication in SciPost.
  • validity: high
  • significance: high
  • originality: high
  • clarity: high
  • formatting: excellent
  • grammar: excellent

Author:  Maxim Breitkreiz  on 2021-07-14  [id 1565]

(in reply to Report 2 on 2021-06-28)

Thank you very much for the positive feedback and the useful comment.

It has been found in previous works that inter-valley scattering can indeed induce different semimetal-insulator transitions
(PRL 115, 246603 (2015)). Those require a small separation of Weyl nodes and/or a large disorder potential.
This effect is thus irrelevant for our work since we consider well-separated Weyl nodes and weak disorder.
Moreover, our focus on long-range disorder and well-separated Weyl nodes makes intra-valley scattering
dominate over inter-valley scattering. For a single Weyl Fermion (and thus only intra-valley scattering)
another type of (perhaps avoided)
critical behavior may occur [PRL 113, 026602 (2014), PRL 121, 215301 (2018), PRB 102, 100201(R) (2020)],
which however becomes relevant only for a vanishing chemical potential (i.e. Fermi level at the Weyl node).

We included a review of the possible disorder-induced phase transitions in the introduction of the new version, 3rd paragraph.

---

## Round 2 · Author Response

Dear Editor,

Thank you for considering our manuscript. We were pleased to find that
both referees recommend publication and have given useful suggestions for minor
changes, which we implemented in the new version of the manuscript and address below.

Sincerely,
Pablo M. Perez-Piskunow, Nicandro Bovenzi, Anton R. Akhmerov, and Maxim Breitkreiz

---

## Round 2 · List of Changes

- introduction section has been rewritten to extend the review of the role of the valley degree of freedom, disorder, and finite-size effects
- added numerical values of particle numbers in the caption of Fig. 4
- added a sentence at the end of section V
- third and second to last paragraphs in the conclusion section have been rewritten
- added 14 new references [5, 7, 18-25, 28, 37-39]
- minor typos have been corrected

---

## Editorial Decision

published